# Novel Approaches to the Establishment of Local Microenvironment from Resorbable Biomaterials in the Brain In Vitro Models

**DOI:** 10.3390/ijms241914709

**Published:** 2023-09-28

**Authors:** Nataliya A. Kolotyeva, Frida N. Gilmiyarova, Anton S. Averchuk, Tatiana I. Baranich, Nataliya A. Rozanova, Maria V. Kukla, Pavel P. Tregub, Alla B. Salmina

**Affiliations:** 1Brain Science Institute, Research Center of Neurology, 125367 Moscow, Russia; 2Department of Fundamental and Clinical Biochemistry with Laboratory Diagnostics, Samara State Medical University, 443099 Samara, Russia; 3Department of Pathophysiology, I.M. Sechenov First Moscow State Medical University, 119991 Moscow, Russia

**Keywords:** brain, metabolism, neurovascular unit, neurogenesis, angiogenesis, biopolymer scaffold, in vitro model

## Abstract

The development of brain in vitro models requires the application of novel biocompatible materials and biopolymers as scaffolds for controllable and effective cell growth and functioning. The “ideal” brain in vitro model should demonstrate the principal features of brain plasticity like synaptic transmission and remodeling, neurogenesis and angiogenesis, and changes in the metabolism associated with the establishment of new intercellular connections. Therefore, the extracellular scaffolds that are helpful in the establishment and maintenance of local microenvironments supporting brain plasticity mechanisms are of critical importance. In this review, we will focus on some carbohydrate metabolites—lactate, pyruvate, oxaloacetate, malate—that greatly contribute to the regulation of cell-to-cell communications and metabolic plasticity of brain cells and on some resorbable biopolymers that may reproduce the local microenvironment enriched in particular cell metabolites.

## 1. Introduction

In recent years, more and more attention has been paid to the study of small molecules that are low-molecular-weight organic substances with molecular mass <1000 Da, metabolic intermediates, regulatory ligands, and signaling molecules able to interact with macromolecules, thereby changing the fluxes in the pathways of intermediate metabolism, or to be released into the extracellular space. Metabolites constitute the majority of cellular molecules and their concentrations are in a wide range. Acting on the metabolism in general, intracellular intermediates participate in a wide variety of molecular transformations and control numerous mechanisms of intercellular interaction [1]. The availability of essential metabolites is ensured by the action of enzymes that regulate energy metabolism, cell redox potential, and total ATP production, thus determining the predominant direction of metabolic reactions in the cell at a given time.

Small molecular weight metabolites affect cell metabolism in different ways by means of their direct interaction with target proteins (enzymes, transporters, and receptors) being located either intracellularly or extracellularly. Small molecules that surround proteins may affect their activity through allosteric regulation [2]. Protein–metabolite interactions may affect proteins with non-enzymatic functions, such as transporters, cytoskeletal proteins, receptors, and transcription factors [3,4,5]. Deciphering the nature and mechanisms of regulatory action of small molecules on cellular proteins would greatly enhance our understanding of the global interactome, metabolome, and exposome [6]. In general, protein-metabolite interactions play a key role in controlling cellular processes due to the establishment of flexible and adaptive modules whose molecular behavior reflects the predominant way of energy production or pattern of signal transduction [7,8]. A detailed study and interpretation of interactions between metabolites and proteins allows for understanding the molecular basis of various (patho)physiological conditions and helps in the development of new approaches to the identification of potential therapeutic targets [9].

The development of an appropriate microenvironment for the long-term in vitro culture of brain cells is still an unresolved question in neurobiology and tissue engineering [10,11]. It relates to the complexity of the extracellular composition in the brain tissue, different metabolic profiles and requirements in neuronal, glial, and endothelial cells, significant effects of developmental aspects, and the variability of current brain tissue in vitro models (isolated cells cultured in a 2D- or 3D-environment, brain-on-chip microfluidic models, organotypic slices, cerebral organoids, etc.). In addition, the brain plasticity phenomenon, which implies numerous tiny changes in the physiology, biochemistry, morphology, and cell-to-cell communications within the brain caused by the action of external stimuli, should be taken into consideration. Indeed, the “ideal” brain in vitro model means the appropriate establishment of the brain plasticity phenomenon in vitro, including dynamic changes in synaptic transmission and remodeling, neurogenesis, and angiogenesis, as well as in the brain tissue metabolism associated with the establishment or remodeling of new intercellular connections. Thus, the extracellular scaffolds that are helpful in the establishment and maintenance of local microenvironment supporting brain plasticity mechanisms are of critical importance.

Overcoming this problem might be achieved with the application of biodegradable (resorbable) biopolymer scaffolds as a “feeding” layer, supporting not only cell growth but also numerous plasticity-associated changes in tissue metabolism and functional activity. In this review, we focus on some carbohydrate metabolites that greatly contribute to the regulation of cell-to-cell communications and metabolic plasticity of brain cells and on biopolymer scaffolds that may reproduce the local environment enriched in particular cell metabolites.

## 2. Small Molecules in the Regulation of Cell Metabolism

The balance between ATP building blocks and substrates is known to be important for the maintenance of metabolic homeostasis. It has been shown that the rate of nutrient uptake correlates with cell size and protein synthesis and is controlled by allosteric regulators. The metabolism of mammalian cells in vitro can vary depending on the availability of nutrients in the growth medium and proceed in a glycolytic or oxidative types of ATP generation [12]. The conversion of pyruvate to lactate in glycolytic cells limits the complete oxidation of glucose to carbon dioxide and water under aerobic conditions, causing a carbon drain from the tricarboxylic acid cycle (TCA) and excessive NAD+ regeneration. This process is known as the Warburg effect. It has been found that most cells can switch between glycolytic and oxidative metabolism as a result of changes in the composition of the nutrient medium or due to significant alterations in their developmental program. In particular, cells grown in low-glucose medium are able to activate the oxidative pathway and thus increase ATP synthesis [13].

*Lactate.* Previously, lactate was considered a byproduct of glucose metabolism and was the focus of controversy for a long time. Nowadays, accumulated data indicate its key role in the regulation of a variety of biological processes, including cell proliferation, differentiation, and cell-to-cell communications [14]. Lactate is known to be a hydroxycarboxylic acid that exists as two stereoisomers—L-lactate and D-lactate. The serum D-lactate content in healthy individuals ranges from 0.013 to 0.2 mM; L-lactate levels are 1–2 mM at rest and rise up to 20–25 mM after exercise [15,16]. It has been shown that lactate is a ligand for catalytic proteins and can alter the conformational stability of glycerol-3-phosphate dehydrogenase [17].

Transport of lactate (as well as pyruvate and ketone bodies) across the plasma membrane is carried out by monocarboxylate transporters (MCT1, MCT2, MCT3, and MCT4). These transporters are encoded by genes of the SLC16 family [18]. The rate of lactate metabolism increases rapidly due to the increased activity of MCTs and is not energy-dependent. It is believed that MCT1 and MCT4 transporters are often overexpressed in actively proliferating cells [19]. In the brain, MCTs are localized on cerebral endothelial cells, perivascular astroglia, neurons, pericytes, and other cells of the neurovascular unit (NVU). They provide lactate multidirectional transport, followed by its uptake into the cells, where lactate is rapidly converted into pyruvate and acetyl-CoA to fuel mitochondrial TCA, or by its accumulation in the extracellular space, where it binds to lactate receptor GPR81 (a G-protein-coupled receptor) whose function in the central nervous system is not well studied [20]. Even the expression of GPR81 in neurons, endothelial cells, and astrocytes has been confirmed [21]. Activation of GPR81 has been shown to occur at lactate concentrations between 0.2 and 1.0 mM [22]. Thus, it was proposed that lactate released form stimulated cells via MCTs acts as a paracrine or autocrine signaling molecule, the “lactormone”. Presumably, brain lactate achieves this lactormone-like regulation of metabolism by interacting with the GPR81 receptor on NVU cells [21,23,24].

There is evidence that alteration of lactate production, transport, and reception in the brain tissue is one of the possible mechanisms in neuroinflammation. In glial cells, it is associated with the expression of inflammasomes that produce proinflammatory cytokines. Chronic activation of Toll-like (TLR) receptors and neuroinflammation leads to the progression of neurodegeneration and brain injury [25,26,27]. GPR81 inhibits NLRP3 inflammasome activity by stimulating the intracellular adaptor protein ARRB2 and impairing NF-κB activation, thereby exerting an anti-inflammatory effect [28,29]. Extracellular lactate acting via GPR81/ARRB2 has been demonstrated to increase HMGB1 acetylation in macrophages by inducing nuclear translocation of p300/CBP acetylase, resulting in increased endothelial permeability [30]. Thus, it is tempting to speculate that elevation of extracellular lactate concentrations might result in either pro- or inflammatory effects in various tissues, including the brain.

Extracellular lactate might also serve as a regulator of intracellular Ca2+ signal transduction [31], cellular energy metabolism, the activity of various channels and transporters, and gene expression [32,33,34]. Lactate has been found to support cognitive functions, learning, and long-term memory formation [35], and may play a neuroprotective role against excitotoxicity [36], and ischemia [37]. However, it is not clear yet what is the relative contribution of GPR81-mediated lactate signaling and MCT-provided uptake of lactate is to its regulatory action in the brain. Thus, deciphering the lactate-driven mechanisms is a promising tool in the development of novel diagnostic and therapeutic approaches in brain pathology [38].

*Pyruvate.* Pyruvate is known to be one of the most important intermediates in metabolism. Pyruvate has a low dissociation constant (pK = 2.49) which indicates its weaker buffering ability compared to lactate (pK = 3.9). Changes in the mitochondrial electron transfer chain activity have been shown to affect the entry of pyruvate into the TCA via pyruvate dehydrogenase, which is a control mechanism for directing metabolites into different metabolic pathways [39]. Like lactate, pyruvate can act as a regulator of protein-protein interactions [40,41].

The entry of pyruvate into the mitochondria requires the activity of the voltage-dependent anion channel (VDAC) located on the outer mitochondrial membrane and a mitochondrial pyruvate carrier (MPC) in the inner mitochondrial membrane. Mitochondrial pyruvate carriers MPC1 and MPC2 regulate the levels of glycolysis and TCA in the cell [42,43]. A number of studies have shown the prognostic role of MPCs in cancer. In particular, MPC1 and MPC2 expressions are associated with favorable clinical outcomes in prostate cancer [44]. Among others, a new signaling pathway KDM5A/MPC-1 was discovered, which promotes the progression of pancreatic cancer by redirecting mitochondrial pyruvate metabolism [45]. However, MPC’s contribution to physiological and pathological conditions in the brain is less known.

By stimulating the flow of the tricarboxylic acid cycle, exogenous pyruvate accelerates lactate oxidation, increases pH, stimulates energy metabolism, and improves mitochondrial function [39]. It has been found that pyruvate is able to activate the hypoxia-induced factor-1 (HIF-1α)-erythropoietin signaling pathway and exert cytoprotective, antioxidant, and anti-inflammatory effects in various pathological conditions [39,46,47]. Pyruvate serves as a major nutrient to support matrix remodeling in a cell culture by inducing the production of α-ketoglutarate, which in turn activates collagen hydroxylation by increasing the activity of the enzyme prolyl-4-hydroxylase [48]. Some studies have shown that the addition of pyruvate induces fibroblast proliferation through NAD+-dependent aspartate synthesis, thereby protecting the cells from aging due to the increased NAD+/NADH ratio [49,50]. Absence of pyruvate in the culture media results in a significant decrease in fibroblast proliferation and activation of fibroblast senescence due to increased β-galactosidase activity [51]. Moreover, pyruvate transamination and NAD+ biosynthesis support the proliferation of cells with succinate dehydrogenase deficiency [52].

Few studies addressed the biological effects of extracellular pyruvate on brain NVU cells or on brain plasticity in general. There are some data on the uptake of pyruvate by astrocytes via MCT1 transporters as a mechanism of mitochondrial fueling [53], and by neurons that can easily convert pyruvate into malate [54]. Thus, due to the shortage of data on the pyruvate’s activity within the NVU, this issue requires further investigation.

*Oxaloacetate.* Oxaloacetate is a critical intermediate in the production of ATP; it should be constantly regenerated for the TCA and mitochondrial electron transfer chain [55]. Simultaneous enhancement of respiratory and glycolytic fluxes increases the cytosolic NAD+/NADH ratio, providing mitochondria with carbon [56]. Oxaloacetate, malonate, and malate are among the inhibitors of succinate dehydrogenase, which is not only an enzyme within the TCA but also an important component in the electron transfer chain. Probably, oxaloacetate has a cytoprotective effect by preventing the backflow of electrons that would otherwise be able to cause increased superoxide release from activated mitochondria [57]. In this context, it may act together with malate via increased expression of superoxide dismutase and glutathione peroxidase [58]. It was shown that oxaloacetate at a final concentration of 0.5–1 μM promotes the thermodynamic stability of glycerol-3-phosphate dehydrogenase, while 16 μM of oxaloacetate causes a decrease in its thermostability [59].

A number of experiments show that oxaloacetate accelerates bone tissue regeneration by promoting proliferation and mineralization. Oxaloacetate-induced regeneration is associated with enhanced proliferation, bioenergetics, and signaling through the BMP/WNT pathway [60]. In silico analysis of oxaloacetate biological activity has shown that it may play a regulatory role in the adaptation of cell metabolism in dermal fibroblasts [61].

The possibility of using oxaloacetate as a therapeutic bioenergetic agent is currently being studied. The antidiabetic effect of oxaloacetate is based on a reliable decrease in the levels of ketone bodies due to suppressed ketogenesis. In the presence of oxaloacetate, acetyl-CoA participates in the formation of citrate and enters the TCA cycle. In the absence of a substrate for this reaction, two acetyl-CoA molecules form an acetoacetate molecule going to the synthesis of ketone bodies [62].

Activation by oxaloacetate of the blood-resident enzyme glutamate-oxaloacetate transaminase, which catalyzes the reversible conversion of oxaloacetate and glutamate into aspartate and α-ketoglutarate, contributes to the reduction of glutamate content in the brain, which prevents neuroinflammation and neurodegeneration. It was observed that the intermediate affects a number of essential functions: it has a general pro-mitochondrial effect by increasing the expression of markers of mitochondrial biogenesis (COX411 and PGC1α), enhances the insulin signaling pathway by phosphorylation of Akt and mTOR, and reduces the concentration of the inflammatory cytokine CCL11. These findings have led to preclinical trials of oxaloacetate in Alzheimer’s disease and ischemic stroke [55,63,64,65,66]. A non-randomized controlled clinical trial has been conducted with oxaloacetate to cure the mental and physical fatigue in patients with myalgic encephalomyelitis/chronic fatigue syndrome (ME/CFS) and long-term fatigue caused by COVID-19. As a result, it was shown that after six weeks of oxaloacetate administration, fatigue symptoms in patients were significantly reduced. Such effects correlate to the increase in glucose uptake due to activation of AMP-activated protein kinase (AMPK), the decrease in lactate production due to inhibition of lactate dehydrogenase, the decrease in reactive oxygen species generation due to an increase in the NAD+/NADH ratio, the suppression of chronic inflammation due to the lower activity of NF-κB, and the prevention of mitochondrial dysfunction due to the activation of PGC1-α-driven mitochondrial biogenesis [67].

*Malate.* As we mentioned above, malate is a redox partner of oxaloacetate acting as a metabolic pathway switch. Malate and oxaloacetate can affect lifespan by activating FOXO/DAF-16 transcription factors and AMPK [68,69]. Inhibition of mammalian mitochondrial aconitase has been shown to enhance the immune response against pathogenic bacteria by modulating the mitochondrial unfolded protein response and oxaloacetate levels in cultured cells [70].

Lactate, pyruvate, oxaloacetate, and malate could be considered major coordinators of glycolysis and mitochondrial oxidative phosphorylation in mammalian cells (Figure 1). Particularly, they are needed for maintaining an adequate cytosolic and mitochondrial NAD+/NADH ratio. NAD+ bioavailability determines the activity of numerous NAD+-converting enzymes, whereas NADH is ultimately required for the electron transport chain in mitochondria. Under the conditions of oxidative or reductive stress, an altered NAD+/NADH ratio might result in aberrant calcium signaling, suppressed DNA replication and DNA repair etc. In glycolysis, pyruvate-to-lactate conversion contributes to the regeneration of NAD+, and the activity of the mitochondrial electron transport chain is associated with NADH oxidation into NAD+. Thus, it is not surprising that the addition of pyruvate and oxaloacetate to cultured cells usually increases the intracellular NAD+/NADH ratio. Particularly, oxaloacetate is rapidly converted into malate, which can cross the mitochondrial membranes, thereby enhancing oxidative phosphorylation and accelerating glycolysis as well. Such effects might be of special importance for actively proliferating cells. Lactate and pyruvate are also able to modulate the energy status of the cell to a certain extent, but a significant increase in their concentration leads to a decrease in the efficiency of glycolysis by a negative feedback mechanism since they are the end products of this metabolic pathway [65,71]. Similar to the fate of pyruvate, the fate of glutamine entering the tricarboxylic acid cycle via α-ketoglutarate is regulated by the ratio of metabolites. A low NADPH+/NADPH ratio and a high α-ketoglutarate/citrate ratio favor reductive carboxylation of α-ketoglutarate to citrate. The action of NADPH-producing succinyl-CoA synthase and nicotinamide nucleotide transhydrogenase provides higher levels of NADPH and stimulates the metabolism of glutamine [39].

*Other small-molecular-weight carbohydrate and lipid metabolites.* Another metabolite of glycolysis is dihydroxyacetone phosphate, which enters the lipid biosynthesis pathway and is converted to glycerol-3-phosphate by the action of glycerophosphate dehydrogenase. However, when dihydroxyacetone phosphate is in excess, it can be converted to methylglyoxal, a toxic compound that interacts with cellular proteins [72]. Accumulation of methylglyoxal in the brain contributes to the deposition of advanced glycation end products, activation of RAGE receptors, progression of aging, and neuroinflammation [73].

The glycerophosphate shuttle is known as a regulator of glycolysis, mitochondrial respiration, and the metabolism of fatty acids. Mitochondrial glycerophosphate dehydrogenase and glycerol-3-phosphate acyltransferase compete for the same substrate (glyceraldehyse-3-phosphate), thereby determining the prevailing direction of cell metabolism at a given time: maintaining the NADH levels, supporting the phospholipid synthesis, or providing the triglyceride synthesis [74]. Indeed, there is evidence that glucose and lipid metabolic pathways converge in the glycerolipid/free fatty acid (GL/FFA) cycle controlled by glycerol-3-phosphate and acyl-CoA. For instance, the lipolytic segment of the GL/FFA cycle plays an important role in the production of signals for insulin secretion, while the lipogenic segment generates signaling molecules that regulate cell survival and death, growth and proliferation [75]. Glycerol-3-phosphate biosynthesis has been shown to contribute to glucose elevation in type 2 diabetes via gluconeogenesis or glyceroneogenesis. Therefore, controlling glycerol-3-phosphate formation in skeletal muscle may be an important therapeutic strategy for type 2 diabetes mellitus [76]. Up-regulation of glycolysis and TCA in activated hippocampal neurons is associated with enhanced biosynthesis of glycerol-3-phosphate [77].

PPAR proteins are expressed in many tissues and organs as regulators of lipid metabolism and homeostasis. Peroxisome proliferator-activated receptor alpha (PPAR-α) interacts with glycerol-3-phosphate, dihydroxyacetone phosphate, and glyceraldehyde-3-phosphate [78]. In Alzheimer’s disease and other neurodegenerative and mental disorders, the expression of PPAR-α and PPAR-γ-1 alpha (PGC-1α) coactivator genes is significantly reduced in the brain. Suppression of PPAR-α may reduce antioxidant and anti-inflammatory mechanisms and may be responsible for alterations in fatty acid transport, lipid metabolism, and altered mitochondrial dynamics in the brain [79].

There are several crossroads that promote carbon storage, conversion, or incorporation into other molecules. For instance, the pentose phosphate pathway (PPP) directs glucose to the production of 5-carbon sugars and the synthesis of nucleic acids and nucleotides [71]. Variations in ATP synthesis in one pathway are compensated by opposite changes in ATP production in other pathways; thereby, metabolic crossroads contribute to all adaptive changes in energy production. Carbohydrate and related lipid metabolites—pyruvate, lactate, oxaloacetate, malate, glycerophosphate, and dihydroxyacetone phosphate—play a major role in maintaining tissue homeostasis (Table 1). Changes in cellular metabolic fluxes affect metabolism in general, which in turn maintains energy production, the NAD(P)/NAD(P)H ratio, the redox state, and the bioavailability of substrates for further catalytic conversion. Therefore, we will further discuss the role of these metabolites in the phenomenon of brain plasticity that could be established and studied in the currently available and prospective brain in vitro models.

## 3. Features of Neurovascular Unit Cell Metabolism and Local Microenvironment in the Regulation of Brain Plasticity

Brain plasticity as a phenomenon based on the adaptation of brain cells and circuits to the action of external stimuli is fundamental for understanding brain development, functioning, and aging. It is commonly accepted that brain adaptation to multiple environmental stimuli includes several complementary mechanisms of plasticity: (i) synaptic plasticity (establishment of new synapses, adjustment of synaptic strength, competitive elimination of synapses) associated with significant changes in neuronal morphology, density of dendritic spines, and expression of immediate early genes; (ii) plasticity within multicellular ensembles (paracrine and autocrine signaling caused by extensive release of neurotransmitters, gliotransmitters, cytokines, and metabolites, as well as astroglial response to neuronal activation, microglia-mediated degradation of synapses, remodeling of neuronal circuits, etc.); (iii) plastic changes in the metabolism and local microcirculation (cooperative activity of mitochondrial oxidative phosphorylation and glycolysis in neuronal and glial cells, local vasodilation and corresponding changes in the permeability of the blood-brain barrier, etc.); (iv) development, maturation, and death of brain cells (recruitment of stem cells and progenitor cells, neurogenesis, gliogenesis, apoptosis, autophagy, (neo)angiogenesis, functional integration of newly born cells into the pre-existing neuronal circuits); (v) induced remodeling of extracellular matrix underlying juxtacrine signaling, adhesion and migration of cells; (vi) controlled permeability of tissue barriers (blood-brain barrier, blood-cerebrospinal fluid barrier, meningeal barrier, etc.); (vii) changes in the passage of tissue fluids (interstitial fluid, cerebrospinal fluid), and glymphatic clearance of the brain tissue [88,89,90,91,92,93].

Metabolic flexibility and even metabolic reprogramming are of utmost importance for brain plasticity. All known plastic events in the brain tissue are associated with transient or long-lasting changes in the metabolism. It should be noted that different cells within the brain neurovascular unit (NVU) are characterized by specific metabolic features, briefly shown below.

Brain stem and progenitor cells. Stem cells and neuronal progenitors use glycolysis and fatty acid oxidation, lipogenesis, and extensive cholesterol metabolism for maintaining their own pool or for their recruitment within neurogenic niches [92,94,95,96]. Along with proliferation and differentiation, immature neuronal cells gradually reduce glycolysis and fatty acid oxidation but activate mitochondrial oxidative phosphorylation [95,96,97,98,99]. However, glycolysis-produced lactate might be much longer required for epigenetic changes (lactylaion of histones) associated with the induction of neuronal differentiation of progenitors [100]. As other actively proliferating cells, neural progenitors in neurogenic niches use glutaminolysis to produce intermediates of the TCA [101], depend on the availability of extracellular glucose for ATP generation and PPP implementation, as well as in lactate supply from extracellular sources for their high mitotic activity [102]. Since extensive glutaminolysis is coupled to mTOR-dependent activation of glycolysis and lactate production [103], cooperative activity of these two metabolic pathways might be required for sufficient proliferation of progenitor cells within neurogenic niches.

Neurons. Mature neuronal cells mainly depend on mitochondria to support extensive synaptic activity and are able to transfer mitochondria along their dendrites to the peri-synaptic area [104,105]; however, some data suggest that neurons may utilize glucose and activate glycolysis for lactate overproduction upon their stimulation [106]. Indeed, upon activation, neurons become more dependent on the availability of lactate, either of neuronal or astroglial origin [107]. Such dependence correlates to the nature of a task: a more demanding task requires lactate, but glucose is needed for fueling neuronal cells in a less complex experience [108]. In general, mitochondrial dysfunction in neurons leads to the induction of TCA anaplerosis; the activity of pyruvate carboxylase and other enzymes responsible for the synthesis of TCA intermediates is greatly elevated in mitochondria-deficient cells for their better survival [109].

Neurons depend on tightly controlled glutamate metabolism: glutamate is released into the synaptic cleft as a major excitatory neurotransmitter, and then it should be taken by astrocytes for the replenishment of glutamate stores in neuronal cells via glutamate-glutamine conversion and glutamine release from astrocytes [110]. Also, glutamate and oxaloacetate serve as intracellular regulators of metabolic pathways; oxaloacetate inhibits complex II in mitochondria, while glutamate attenuates this inhibition and stimulates reactive oxygen species production in neuronal cells [111]. Antioxidant activity in neurons is mainly provided by elevated PPP-mediated generation of NADPH; therefore, glycolytic activity in neuronal cells is low due to continuous proteasomal degradation of 6-phosphofructo-2-kinase/fructose-2, 6-bisphosphatase-3, and as a result, glucose is guided to the PPP instead of glycolysis [112].

Astrocytes. Astroglial cells store glycogen and activate glycolysis for rapid production and release of lactate (even in aerobic conditions) as an alternative fuel for neighboring neurons in the mechanism known as neuron-astrocyte metabolic coupling [113]. They produce and utilize fatty acids to support the high demand for neurons in fatty acids and lipids [114,115], or to prevent the extensive reactive oxygen species-generating activity of the electron transport chain in their own mitochondria [116,117,118].

Astrocytes are equipped with enzymes of glutaminolysis, glutamate conversion into glutamine, and complete oxidation of glutamate via the ”pyruvate recycling pathway”, glycogen synthesis and degradation, synthesis of all components of the TCA, and de novo synthesis of glutamate and glutamine from glucose [111,119,120,121]. Comparing with neuronal cells, astrocytes demonstrate much higher activity of PPP, which is needed for the generation of NADPH and replenishment of the reduced glutathione pool, which is further supplied to neuronal cells for protection from oxidative stress [113]. Interestingly, in Drosophila, glucose could be transported from insulin-stimulated astrocytes to neurons to support the neuronal PPP, and this mechanism is ultimately required for long-term memory [122].

In contrast to neurons, astrocytes possess glutamine synthase-mediated synthesis of glutamine as well as pyruvate carboxylase-mediated conversion of pyruvate into oxaloacetate, which can be further converted into alpha-ketoglutarate and glutamate [123]. The rate of pyruvate carboxylase activity is different in various brain regions: it is lowest in the cerebellum, intermediate in the cortex and hippocampus, and highest in the striatum of awake rats, which is correlated with the rates of glutamate cycling and total glucose oxidation [124]. Like in neuronal cells, there is a coordination of mitochondrial and extramitochondrial energy production in astrocytes: AMPK is activated in conditions of ATP shortage then glycolysis and pyruvate carboxylase-stimulated TCA anaplerosis, as well as glutamate oxidation via pyruvate recycling, are augmented in astrocytes [125].

The glutamate-glutamine cycle is essential for the appropriate activity of neurons and astrocytes within the tripartite synapses [110]. Impairment of this machinery is associated with cognitive dysfunction, as shown in iPSC-derived cells obtained from patients with fronto-temporal dementia. Particularly, they demonstrate neuronal glutamine hypermetabolism, decreased intracellular glutamine content, excessive production of mitochondrial reactive oxygen species, glucose hypometabolism, and altered glycolysis, and the development of reductive stress associated with a higher NADH/NAD+ ratio in the mitochondrial matrix vs. increased astroglial glutamate uptake and metabolism [126].

Microglia. Microglial cells change their phenotype from mitochondrial production of ATP into a glycolytic one upon stimulation and polarization [127]. This mechanism includes lactate-driven lactylation of histones and epigenetic changes associated with the pro-inflammatory phenotype of microglia [128]. However, the neuroprotective action of microglia might require more efficient mitochondrial oxidative phosphorylation [129]. Microglia efficiently metabolizes fatty acids via beta-oxidation, and such metabolic activity of microglia is needed for brain plasticity and memory consolidation [130]. In general, saturated fatty acids support the pro-inflammatory phenotype of microglia, while unsaturated fatty acids promote the anti-inflammatory phenotype [131]. PPP is important for NADPH generation in activated microglial cells (as a cofactor for reactive oxygen species-generating NADPH oxidase in phagocytosis); therefore, it is not surprising that chronic neuroinflammation in Parkinson’s disease is marked by elevated PPP activity in midbrain microglia and excessive degeneration of dopaminergic neurons [132].

In conditions of reduced glucose availability, microglia may utilize glutamine in glutaminolysis for the generation of alpha-ketoglutarate and TCA fueling [133]. Glutaminolysis, where glutamine is converted into glutamate, aspartate, pyruvate, lactate, alanine, and citrate, is often considered a major metabolic pathway for energy generation in actively proliferating cells and is known as glutaminolysis-induced metabolic reprogramming [103,134], but recently it became clear that glutaminolysis in the brain attributes to the activity of microglial cells and supports their metabolic flexibility needed for migration and patrolling functions [133].

It should be noted that glutaminase, which is a limiting mitochondrial enzyme of glutaminolysis (it converts glutamine into glutamate in a reaction that also yields ammonia), is expressed in neurons [135] and—at very low levels—in astrocytes [136]. Since glutamine-derived metabolites promote numerous secondary metabolic reactions (e.g., citrate is used for fatty acid synthesis, oxaloacetate and aspartate go to the nucleotide synthesis and NADPH generation) [103], whereas glutamine itself is a nitrogen donor for the synthesis of purine and pyrimidine nucleotides [137] and a precursor of non-essential amino acids (glutamate, asparagine) [103], one should consider glutamine-linked metabolism as an important mechanism of metabolic plasticity in NVU cells.

Oligodendroglia. Oligodendrocytes use mitochondria not only for ATP production but also for myelin synthesis [138] and utilize glycolysis at the final steps of axonal myelination. Moreover, they consume lactate in physiological conditions [106] or provide glycolysis-derived lactate and pyruvate for neighboring neuronal axons [139,140]. Oligodendroglia appears as not only a source of lactate in the brain but also as a target for lactate-mediated signaling action: extracellular lactate stimulates consumption of glucose and pyruvate metabolism but does not affect glycolysis in oligodendrocytes [123]. Moreover, extracellular (astroglia-released) lactate promotes differentiation of oligodendrocytes, particularly when glucose is less available [141]. Like astroglial cells supporting the metabolic requirements of neuronal synapses, oligodendrocytes are able to support the energy needs of neuronal axons and facilitate mitochondrial ATP production there in a NAD+/sirtuin-dependent [142] or lactate-dependent [143] manner. Consumption of ketone bodies is evident in proliferating oligodendrocyte precursor cells [144] that also depend on mitochondrial oxidative phosphorylation [145]. Oligodendrocytes demonstrate high activity of TCA and PPP: mature oligodendrocytes may use not less than 10–15% of glucose in the PPP compared with glycolysis [123], but overall glucose consumption correlates to prominent synthesis of lipids in oligodendroglia [146]. In sum, the production of lactate/pyruvate in glycolysis and NADPH in the PPP supports extensive lipid metabolism in oligodendroglia: the synthesis of myelin from acetyl-CoA [141] and the synthesis of neurosteroids [147].

Endothelial cells. Brain microvascular endothelial cells (BMECs) contributing to the establishment of the blood-brain barrier (BBB) are characterized by a high density of mitochondria, their ability to transfer mitochondria to other cells if needed, as well as by a high dependence on lactate availability in the extracellular space [143]. Extracellular lactate, which is transferred between the NVU cells via MCTs or acts at plasma membrane GPR81 receptors, supports mitochondrial biogenesis and the maturation of mitochondria, thereby stimulating barriergenesis and angiogenesis that accompany almost all plastic changes in the brain [148,149]. Vascular senescence seen in neurodegenerative diseases (e.g., in Alzheimer’s and Parkinson’s type neurodegeneration) or in physiological aging is often associated with significant changes in the metabolic preferences of primary human brain microvessel endothelial cells cultured in vitro. Particularly, glycolysis is a preferable pathway for ATP synthesis and glutamine is a preferable fuel for mitochondrial oxidative phosphorylation in young (7–9 passages) endothelial cells; similar contribution to ATP production of glycolysis and mitochondrial oxidative phosphorylation, and equal utilization of glutamine, glucose, and fatty acids as fuels for mitochondria are evident in pre-senescent brain endothelial cells (13–15 passages); significant decline in mitochondrial respiration and glycolysis is seen in senescent endothelial cells (20–21 passages) [150]. The same tendencies have been clearly demonstrated in BMECs isolated from young (2–6 months) and aged (20–22 months) male mice [151,152].

Endothelial cells take part in the establishment of a new microvascular network either in embryogenesis (developmental vasculogenesis and angiogenesis) or in the adult brain (experience- and damage-driven angiogenesis). These mechanisms are based on the activity of endothelial progenitor cells, migrating non-proliferating tip-type endothelial cells, and proliferating stalk-type endothelial cells in the tissue [153]. Uptake of extracellular glucose for energy production is indispensable for angiogenesis and barriergenesis; thereby, blocking glucose transport in brain endothelial cells disrupts postnatal angiogenesis and the establishment of the barrier [154]. Endothelial progenitor cells demonstrate immature mitochondrial morphology but possess high glycolytic activity and produce lactate for the establishment of a pro-angiogenic microenvironment [143]. Endothelial progenitor cells are characterized by extensive glycolysis, lactate production, and high activity of NAD+-producing and NAD+-consuming enzymes [155]. Tip cells have higher glycolytic activity than stalk-type endothelial cells, and overexpression of 6-phosphofructo-2-kinase/fructose-2, and 6-bisphosphatase-3 results in better angiogenic properties of tip cells [156]. However, there are some contradictory data: tip-type HUVEC cells might have lower glycolytic activity and higher mitochondrial respiration than non-tip ones, whereas stalk cells require mitochondrial oxidative phosphorylation to support their proliferative activity [157]. Glutamine-dependent synthesis of asparagine is indispensable for the growth and sprouting of new microvessels: tip endothelial cells use glutamine for the synthesis of asparagine, which is further used for protein synthesis and might affect mTOR signaling [158]. Expression of glutaminase in endothelial cells determines their competitiveness in getting the tip phenotype. Even though the activity of this enzyme does not affect NO-synthesizing activity or the expression of adhesion proteins, glutamine deprivation does not cause an energy crisis since the glycolytic activity of endothelial cells is still rather high [158]. Higher dependence on glutamine oxidation and glucose oxidation was found in tip endothelial cells (HUVEC) than in non-tip cells, whereas stalk endothelial cells demonstrated higher activity of fatty acid oxidation [157]. Whether the same metabolic features are applicable to BMECs that are intrinsically different from HUVEC and other endothelial cells (e.g., due to the higher density of mitochondria) requires further assessment.

In sum, the local microenvironment and activation of various NVU cells significantly affect their metabolism, leading to metabolic reprogramming and changes in the relative contribution of different metabolic pathways to the total energy production or synthesis of biomolecules. As we have discussed before in detail [97,148,159], the metabolism of NVU cells provides the controlling microenvironment several aspects of brain plasticity. Particularly, lactate triggers synapse-specific long-term potentiation (LTP) in CA3 hippocampal neurons in vitro [160], extracellular lactate levels in the hippocampus increase during the development of LTP and memory consolidation in vivo [161], and acute lactate rise promotes hippocampal neurogenesis in the rodent adult brain in a MCT2-dependent but not in a GPR81-dependent manner [162]. Lactate demonstrates neuroprotective properties and regulates mitochondrial quality control mechanisms in brain cells [163,164]. Impairment of glutaminase activity in the brain results in altered hippocampal excitability [165]. However, overexpression of glutaminase is associated with the progression of neuroinflammation, synaptic dysfunction, and learning deficits in vivo [166]. Chronic administration of oxaloacetate improves neurogenesis and angiogenesis and stimulates mitochondrial biogenesis in the hippocampus of mice [167]. All these data suggest that manipulation with the availability of low molecular weight metabolites in the extracellular microenvironment is a potent tool for controlling the establishment and functional maturation of the brain in vitro cultures.

## 4. Biodegradable Biopolymers as a Source of Small Molecules in the Tissue In Vitro Models

The majority of in vitro tissue models face the problem of inappropriate reconstruction of cell metabolism: it is either inefficient or suboptimal because the nutrients supplied in the media might not be efficiently consumed or their utilization could lead to the accumulation of toxic by-products, intermediates, and toxic metabolites, especially in static cultures [12]. As well-known examples, methylglyoxal and lactate as glycolysis side product and main product, respectively, show toxic effects when they are significantly accumulated in the extracellular space [12,168]. Ammonia accumulates in the extracellular medium of cultured cells due to the imbalanced production of TCA intermediates caused by excessive catabolism of amino acids or glutamine breakdown [12,168]. Thus, integration of multiplex-sensitive sensors to detect metabolite levels, functional activity, and viability of cells and processing of real-time data are needed for the optimization of the microenvironment established in long-lived cell and tissue in vitro models [169].

Partially, this problem could be solved with the application of scaffolds that might support cell development, movement, intercellular interactions, and balanced metabolism. Various physical and chemical properties of (bio)scaffolds that are currently used or tested for the brain tissues grown in vitro have been extensively reviewed elsewhere [10,170,171,172,173,174]. Natural polymers such as collagen, gelatin, alginate, and fibrin are often used as scaffolds in tissue engineering, but they do not release significant amounts of low-molecular-weight metabolites upon degradation; therefore, they can’t be considered a reliable source of cytokine-like molecules directly affecting the metabolic plasticity of NVU/BBB cells in vitro. Thus, we focus on the development and application of biodegradable polymers that control the availability of the metabolites discussed above. Managing the composition of extracellular medium with biopolymer scaffolds allowing the release of some glycolytic or mitochondrial metabolites is a prospective way to optimize the culture conditions, especially for NVU or BBB in vitro models consisting of various populations of functionally interacting cells.

Polylactones such as poly(lactic acid) (PLA), poly(glycolic acid) (PGA), polycaprolactone (PCL), and their copolymers, poly(lactic-co-glycolic acid) (PLGA), possess good biocompatibility with the cells of the central and peripheral nervous systems, thereby becoming the most commonly used synthetic biodegradable compounds whose biomedical applications are focused on nerve tissue engineering and in vitro brain modeling. PLA and PGA are thermoplastic polymers characterized by polyester links of lactic acid or glycolic acid, respectively.

The improvement of synthetic polymers using surface modification techniques and the addition of growth factors to the material have expanded the use of scaffolds as carriers for targeted, controlled delivery of drugs, peptides, proteins, monoclonal antibodies, growth factors, DNA, and RNA [175,176,177]. Reactive micelles engineered from polylactides have been developed and used as mRNA delivery vectors [178]. In addition, allylic cationic polylactides (CPLA) have been developed, which are effective for siRNA and pDNA delivery to cancer cells and are used in gene therapy [179].

PLA and PGA have low toxicity and can be absorbed or hydrolyzed by growing cells. PLGA has a lower melting point. By increasing the glycolide content, the monomer concentration can control the degradation rate of this biopolymer. PGA, PLA, and PLGA support the growth of NVU cells by determining the interactions between the cells and the scaffold. PLA has a hydrophobic surface, which affects the amount of absorbed protein and cell adhesion. Through surface modification of the scaffold, neuronal cell behavior—differentiation, migration, and axon guidance—can be controlled [175,180,181]. The hydrolysis of PLA produces L-lactic acid; thereby, the extracellular action of released lactate via GPR81 receptors or its cellular uptake via MCTs might significantly affect the ATP production and physiology of cultured cells (Figure 2).

PLA can maintain mechanical and structural integrity and provides adequate mechanical properties for long-term regenerative processes. Thus, PLA-based biomaterials can be designed as scaffolds to mimic the brain extracellular matrix. There are a number of experimental studies showing that alteration of the biopolymer surface due to biochemical or physical processes increases its hydrophilicity, thereby affecting the growth and branching of neurites in neuronal cultures [182,183,184].

Poly(lactic-co-glycolic acid) (PLGA), an aliphatic complex polyester, is a copolymer of poly(lactic acid) (PLA) and poly(glycolic acid) (PGA) with high biocompatibility and low toxicity. When degraded, it forms the hydrolysis products lactate and glycolate that affect TCA and gluconeogenesis. The possibility of varying the ratio of lactic and glycolic acid monomers, crystallinity, and molecular weight has led to different types of PLGAs [185]. In particular, increasing the ratio of glycolic acid links to lactic acid links in PLGA increases the degradation rate; increasing the molecular weight and crystallinity of the structure decreases the degradation rate [186]. PLGA also might be considered a source of long-term released lactate, whose further action on cultured cells would require its uptake via MCTs or action at GPR81 receptors.

However, currently, PLGA is mainly used to construct a wide range of nanostructures, including polymeric micelles, nanogels, dendrimers, nanocapsules, nanorods, and theranostic agents. In particular, PLGA with terminal acidic groups allows easy conjugation with drugs with nucleophilic functionality [187]. PLGA nanostructures have demonstrated the ability to penetrate the BBB for delivery of antitumor drugs [188,189,190,191], and genetic constructs [192]. Modified PLGA nanospheres can serve as therapeutic gene delivery vehicles for spinal cord injury. In a model of spinal cord injury in rats, PLGA/DC-Chol nanospheres loaded with the vascular endothelial growth factor gene enhanced angiogenesis at the site of injury, improved tissue regeneration, and led to better recovery of motor function [193]. PLGA nanoparticles loaded with triptolide attenuate the damage of the cell model of Alzheimer’s disease when administered intranasally, which is one of the approaches to deliver therapeutic agents via the olfactory pathway bypassing the BBB [194].

PLGA nanoparticles modified with poly (ethylene glycol) (PEG) are used to reduce protein adsorption and limit nonspecific cellular uptake. The inclusion of surface-active agent molecules such as cholic acid (CHA), F68, F127, P80, and polyvinyl alcohol (PVA) modifies the ability of the nanoparticles to diffuse into the brain tissue. In particular, after in vivo administration, PLGA-PEG/P80 nanoparticles demonstrated enhanced penetration through the BBB and subsequent internalization in neurons and microglia [195].

Multichannel PLGA scaffolds seeded with Schwann cells and bone marrow-derived mesenchymal stem cells (MSC) have been shown to have a positive effect on neuronal regeneration for peripheral nerve repair in tissue-engineered nerve transplants (TENGS) [196,197].

As we mentioned above, the establishment of cerebral microvessels is a key component of brain plasticity. It might be coupled to extensive synaptic remodeling in active brain regions and the induction of neurogenesis in neurogenic niches. One of the approaches for creating bioartificial vessels involves the design and fabrication of cell-free microcapillary equivalents from novel biopolymeric materials. PLA, PLGA, and PCL are used as composites to create vascular prostheses. Some can be combined with natural polymers to improve cell attachment and infiltration, while others are a combination of synthetic polymers to mimic multi-layer tubular patterns similar to blood vessels [198]. Thus, it is tempting to speculate that the establishment of functionally competent cerebral microvessels in the brain in vitro models might be achieved with PLA-, PLGA-, or PCL-based biopolymer scaffolds. Particularly, it might be applicable for the artificial vascularization of iPSC-derived cerebral organoids whose growth and development in vitro are altered because of the absence of microvessels. Also, it could be reasonable to use these biopolymers for the establishment of vascular scaffolds needed for neurogenic niches in vitro models where the development of neural stem and progenitor cells is directly controlled by local microvessels.

The poly(L-lactic acid) nanofiber mesh framework demonstrates a significantly better proliferation rate of endothelial cells on vascular prostheses [199]. The behavior of endothelial cells was shown to be regulated by the dynamically changing microenvironment of the biodegradable PLA polymer [200]. A novel approach has been developed to generate the internal topology of branched three-dimensional networks of microchannels with rounded cross sections composed of biocompatible, degradable PLA substrates. This technique allows for the successful culture of endothelial cells in the branched microvascular networks [201]. Moreover, 3D poly(lactic acid) scaffolds have been shown to promote different behaviors of endothelial progenitors and adipose tissue-derived stromal cells compared with 2D cultures [202]. These results highlight the importance of studying cellular behavior depending on the architecture of the matrix and the type of biomaterial used.

In sum, the application of poly(L-lactic acid)-based scaffolds for the culture of BMECs and vascularization of the brain tissue in vitro is highly reasonable: lactate released from the polymer may promote angiogenesis either due to action on lactate receptors expressed in BMECs, or by supporting mitochondrial oxidative phosphorylation required for stalk-type cells proliferation and sprouting angiogenesis.

Poly(ethylene glycol) and poly(trimethylene carbonate) (PTMC) are synthetic polymers that have been studied as potent scaffolds for regeneration and targeted drug delivery in the CNS. Ultrafast degradable tyrosine-based polymers containing PEG and PTMC have been developed. Such polymers might be suitable for application in neural prostheses and neural interfaces [203].

Poly(β-L-malic acid) (PMA) is a natural biopolymer composed of repeating L-malyl links that are linked by ester linkages that form an L-malate molecule. PMA and its derivatives possess the properties of biocompatibility, biodegradability, water solubility, and non-immunogenicity and may have great potential applications as drug delivery systems and in biomaterial manufacturing. In addition, L-malic acid, a hydrolysis product of PMA, can enter the TCA cycle as a source of energy and carbon for amino acid biosynthesis and cell growth [204,205,206]. Poly(benzyl malate) (PMLABe), poly(hexyl malate) (PMLAHe), and poly(malic acid-co-benzyl malate) (PMLAH/He) nanoparticles showed no cytotoxicity in the in vitro culture of the J774 macrophage cell line, had no toxic effects on mice, and stimulated bone repair and wound regeneration [207,208]. In addition, PMA was used as a platform for the synthesis of nanosized delivery systems [209]. Given the high density of carboxyl groups in the side chains, PMA has high water solubility and chemical reactivity. Chemically, the carboxyl groups of the PMA side chain allow them to efficiently link to many functional groups of bioactive molecules, including ligands, chemotherapeutic drugs, imaging agents, and therapeutic antibodies. These properties make PMA a good candidate for use as a targeted drug delivery system [185,205,210,211]. It has been shown that the higher the molecular weight of PMA, the higher the strength of PMA-based nanoconjugates and nanoparticles [209]. Based on PMA, new imaging and therapeutic BBB crossing agents have been designed as novel diagnostic and therapeutic tools in brain tumors and in Alzheimer’s disease [212,213]. It was shown that PMA-based nanoconjugates provide a multifunctional delivery system that can efficiently pass through the BBB, accumulate in brain tumors, thereby allowing visualization and release of drugs to inhibit angiogenesis and tumor growth [211,214].

Another biodegradable polymer is poly(l-lactide-co-β-malic acid) (PLMA) with side carboxyl groups. Its degradation products are non-toxic lactic acid and malic acid, which are metabolic intermediates [215,216]. PLMA is used in tissue engineering, cell labeling, and targeted drug delivery [217]. Furthermore, in vitro biocompatibility evaluation has confirmed the great potential of PLMA biomorphic matrices for bone tissue [216,218,219].

Poly((R,S)-3,3-dimethylmalic acid) (PDMMLA) is a biopolymer belonging to the poly (malic acid) family. PDMMLA is copolymerized with PLA to improve cell attachment, proliferation, spreading, and prevent platelet aggregation; it also has a good affinity for HUVEC. The hydrolysis of amorphous PDMMLA is a natural and non-cytotoxic product that participates in the pantothenate biosynthesis pathway. The physicochemical and mechanical properties of this polymer allow it to be used for the coating of endovascular stents [220,221]. However, the application of PTMC, PMA, PLMA, and PDVVLA to the brain in vitro long-term models needs further assessment.

Table 2 summarizes current data on the key properties of biopolymer scaffolds that could be considered permanent sources for the brain cells-controlled release of carbohydrate metabolites with well-established regulatory functions. The application of these biopolymers in the brain tissue in vitro modeling requires extensive experimental studies.

## 5. Conclusions

The development and application of novel biopolymer scaffolds in the brain in vitro models (2D models, 3D models, cerebral organoids, and brain-on-chip microphysiological systems) should take into consideration the possibility of controlling the brain plasticity phenomenon by providing the release of small-molecular-weight metabolites into the culture medium. This is in agreement with the higher diversity of metabolic features of various brain cells within functionally competent tissue compartments like the neurovascular unit, neurogenic niche, cortex minicolumns, angiogenic niche, etc. We believe that biopolymer-resorbable scaffolds releasing lactate and other carbohydrate metabolites are of great interest and importance.

## 6. Future Directions

Further progress in the establishment of brain in vitro models resembling plastic changes underlying the complex behavior of brain cells (e.g., synaptic communication, synaptogenesis, synapse elimination, neuritogenesis, myelination, neurogenesis, angiogenesis, neural circuits remodeling, etc.) depends on the development of novel biomaterials with high biocompatibility and low toxicity. Such materials could not only mimic the extracellular matrix composition but also serve as a controllable source of metabolites (e.g., lactate, pyruvate, oxaloacetate, malate, etc.) whose uptake or receptor-mediated action in cultured cells would result in changes in cell metabolism and functional activity. Thus, we need in the novel generation of biopolymer scaffolds supporting the establishment of local microenvironment with the desired action (e.g., promoting or suppressing neurogenesis, microglial activation, angiogenesis, or neuron-astrocyte metabolic coupling) for the development of appropriate conditions for the brain tissue in vitro models reproducing complex events underlying brain development and experience-driven plasticity.

## Figures and Tables

**Figure 1 ijms-24-14709-f001:**
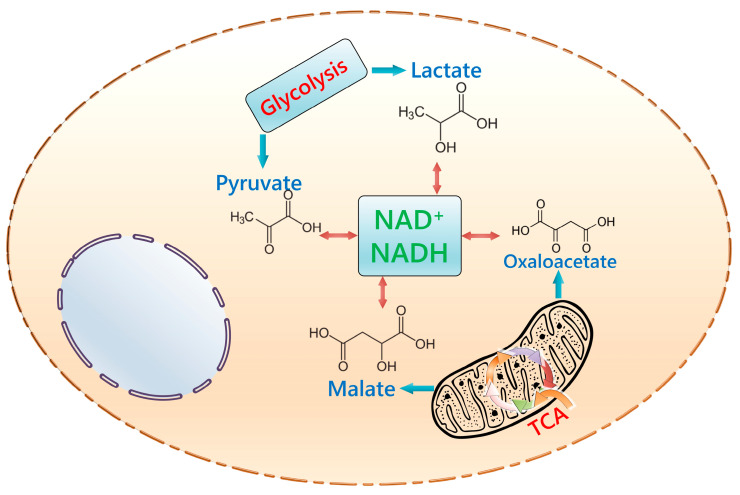
Lactate/pyruvate and oxaloacetate/malate as regulators of NAD+/NADH ratio in mammalian cells. TCA—tricarboxylic acid cycle.

**Figure 2 ijms-24-14709-f002:**
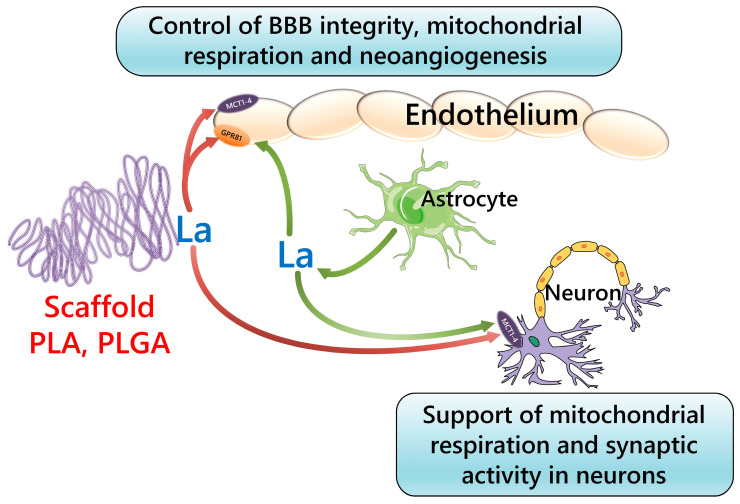
Putative effect of hydrolysis of PLA/PLGA bioscaffolds on cell physiology in vitro. PLA—polylactic acid; PLGA—polylactic-co-glycolic acid; GPR81—G-protein-coupled lactate receptor; MCT1-4—monocarboxylate transporters (MCT1, MCT2, MCT3 and MCT4); La—lactate; BBB—blood-brain barrier.

**Table 1 ijms-24-14709-t001:** Key carbohydrate metabolites with signaling and regulatory properties: synthesis, transport, reception, and major biological effects.

	Metabolites			
	Lactate (La)	Pyruvate (Py)	Oxaloacetate (Oa)	Malate (Ma)
Synthesis pathways	LDH Reaction (Py↔La) in Glycolysis	From Glucose in Glycolysis Glucogenic amino acids LDH Reaction (La↔Py) ALT Reaction (Alanine↔Py) NAD-malic enzyme Reaction(Ma↔Py) Amino acids Ala, Ser, Gly, Cys are converted into Py	MDH Reaction (Ma→Oa) in the citric acid cycle Pyruvate carboxylase Reaction (Py→Oa) Phosphoenolpyruvate carboxylase Reaction (in plants and bacteria) (PEP→Oa)	MDH Reaction (Ma→Oa) NAD-malic enzyme Reaction (Py↔Ma)
Impact on other metabolic pathways	Oxidation to Py, then to TCA cycle (energy metabolism) Gluconeogenesis (carbohydrates synthesis) Cori Cycle Glycogen stores	TCA cycle (energy metabolism) Gluconeogenesis (carbohydrates synthesis) Amino acid synthesis (Alanine) Fatty acids synthesis (Py→Acetyl-CoA) Converted to acetaldehyde (ethanol synthesis) La synthesis, LDH Reaction(Py↔La) Oa synthesis	TCA cycle (energy metabolism) Gluconeogenesis (carbohydrates synthesis) Amino acid synthesis (Aspartate) Fatty acid synthesis (Py→Acetyl-CoA) Urea cycle Glyoxylate cycle	TCA cycle (energy metabolism) Fatty acid synthesis (via citrate-malate shuttle) Carbon fixation process (in plants) Amino acid synthesis
Transporters	MCT1-4 [18] La/Py shuttle [77]	MPC1, MPC2 VDAC channels [42,43] MCT2, MCT3 [18] La/Py shuttle [80]	Ma/Oa shuttle [80] Oa/Ma transporter (OMT1 or DiT1) (in chloroplast) [81]	SLC Proteins (SLC25A11, SLC25A1) [82] Ma/Oa shuttle [80,83] Ma/aspartate shuttle [80,83,84] Citrate/malate shuttle [80,83] OMT1 or DiT1 (in chloroplast) [81]
Receptors	GPR81 [85]	GPR31 [86,87]		

**Table 2 ijms-24-14709-t002:** Biodegradable biopolymers.

	Characterization	Production Technologies, Application	Peculiarities	References
Poly(lactic acid) PLA	MP 170–180 °C; TS 60–70 MPa; M 2–4 Gpa; TG 60–65 °C; E 2–6%; Mechanical strength about 4.8 GPa; Density 1.0 L/g; Degradation time 40 weeks in vitro, 30 weeks in vivo; Monomer—lactic acid	− Electrospinning − Melt-blown technique, − 3D printing − Additive manufacturing − Phase separation − Tissue repair and support strategies (biomaterial) − Drug/growth-factor-loaded systems − Scaffold processing for Tissue Engineering	Hydrophobic polymer of a synthetic nature, limits biological signaling and protein absorptionPLA scaffolds are unstable in size/structure, often breaking and wrinkling	[175,222]
Poly(lactic-co-glycolic acid) (PLGA)	Permeability, swelling, deformation, degradation rate, can be controlled by altering the ratio of PLA:PGATG 45–55 °C; Monomers—glycolic and lactic acids	− Electrospinning − Melt-blown technique, − 3D printing − Tissue repair and support strategies (biomaterial) − Drug/growth-factor-loaded systems − Scaffold processing for Tissue Engineering	PLGA has great strength and biodegradability. It is limited by their hydrophobicity and lack of cell recognition sites	[175,177,185]
Poly(glycolic acid) (PGA)	TG 35–40 °C; MP 225–230 °C; M 7 GPa; Crystallinity 45–55%; Monomer—glycolic acid	− Drug/growth-factor-loaded systems − Scaffold processing for Tissue Engineering − Textile technologies in the form of non-woven felts	PGAs gradually lose strength 1–2 months after implantation. PGA-based nanowires are limited to bridging a small nerve gap. Fibers of PGA exhibit high strength and are particularly stiff. The degradation product, glycolic acid, is nontoxic, but it is metabolized to oxalic acid, which could make it dangerous	[223,224]
β-Poly(malic acid) (PMA), poly(1,8-octanediol malic acid) (POM)	TS: 7–25 MPa; Elongation 3–14% Compressive stress 6–14 kPa; Compressive Young’s modulus 0.12–0.25 kPa; Degradation in PBS at 37 °C from 2 to 13 weeks; Monomer—malic acid	− Solvent casting technique − The particulate-leaching technique − Drug/growth-factor-loaded systems − Scaffold processing for Tissue Engineering	Surface chemistry and hydrophilicity/hydrophobicity may control cell attachment and subsequent cell proliferation A highly hydrophilic surface with large amounts of polar groups such as carboxyl and hydroxyl groups may impair cell adhesion POM and its degradation products are not toxic, biocompatible in vivo	[204,205,208,216]
poly(l-lactide-co-β-malic acid) (PLMA)	TG 49 °C; porosity of 85.5%; compressive modulus 766 ± 36 kPa; Monomers—malic and lactic acids. The degradation was accelerated upon heating	− Drug/growth-factor-loaded systems − Scaffold processing for Tissue Engineering	High capability for cell uptake and for their stability in the blood stream	[216,218,219]

Melting point—MP; Tensile strength—TS; Glass transition temperature—TG; Modulus—M; Elongation—E.

## Data Availability

Not applicable.

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
