# Peer review of "Novel Approaches to the Establishment of Local Microenvironment from Resorbable Biomaterials in the Brain In Vitro Models"

_ijms, 2023, doi:10.3390/ijms241914709_

Round 1

Reviewer 1 Report

Overall comments:

The subject matter of this paper deals with the recent advance on the establishment of local microenvironment from resorbable biomaterials in the brain in vitro models. The paper discusses the need for novel biocompatible materials and biopolymers as scaffolds for brain in vitro models that can mimic brain plasticity mechanisms, such as synaptic transmission, neurogenesis, and angiogenesis. It focuses on carbohydrate metabolites (lactate, pyruvate, oxaloacetate, malate) that regulate cell-to-cell communication and metabolic plasticity, as well as resorbable biopolymers that can recreate the local environment enriched in these metabolites.

The manuscript itself is considered to be theoretically and structurally reasonable. This kind of review is well worth investigating, and the authors did well in stating what the goal of the paper is. However, there are some specific concerns that should be clearly addressed. If such issues were all cleared by the authors, this paper seems to be qualified to secure its publication.

Specific concerns:

1. I recommend to suggest a schematic image that can convey the entire context of this review.

2. In paragraph 1, the research showing that the addition of pyruvate induces the proliferation of fibroblasts (page 4, line 163-167) and that oxaloacetic acid promotes bone tissue regeneration (page 4, line 184-189) seems to have been taken out of context. It is more appropriate to present the results of the paper on what mechanisms these substances act in relation to nerve cells.

3. In paragraph 4, PLA, PGA, PCL, and PLGA were demonstrated as biodegradable polymers used to fabricate a scaffold that supports appropriate cell metabolism. These are all synthetic polymers, but it is necessary to summarize examples using natural polymers such as collagen/gelatin, alginate, and fibrin.

4. Representative images (Figures from references or the authors’ own) should be added to each section.

none

Author Response

Reviewer # 1:

The subject matter of this paper deals with the recent advance on the establishment of local microenvironment from resorbable biomaterials in the brain in vitro models. The paper discusses the need for novel biocompatible materials and biopolymers as scaffolds for brain in vitro models that can mimic brain plasticity mechanisms, such as synaptic transmission, neurogenesis, and angiogenesis. It focuses on carbohydrate metabolites (lactate, pyruvate, oxaloacetate, malate) that regulate cell-to-cell communication and metabolic plasticity, as well as resorbable biopolymers that can recreate the local environment enriched in these metabolites.

The manuscript itself is considered to be theoretically and structurally reasonable. This kind of review is well worth investigating, and the authors did well in stating what the goal of the paper is. However, there are some specific concerns that should be clearly addressed. If such issues were all cleared by the authors, this paper seems to be qualified to secure its publication.

 Authors: Dear Reviewer! We are grateful for the interest to our manuscript. According to the Reviewer’s suggestion, we revised the manuscript (all the corrections are indicated with yellow).

Reviewer # 1:

  1. I recommend to suggest a schematic image that can convey the entire context of this review.

Authors: We added a schematic image to the manuscript (in the attached file).

Reviewer # 1:

  1. In paragraph 1, the research showing that the addition of pyruvate induces the proliferation of fibroblasts (page 4, line 163-167) and that oxaloacetic acid promotes bone tissue regeneration (page 4, line 184-189) seems to have been taken out of context. It is more appropriate to present the results of the paper on what mechanisms these substances act in relation to nerve cells.

Authors: We corrected this part accordingly, added some details throw out the section 2. We added information about pyruvate: page 4, line 174-178; oxaloacetate p.4, line 187, p.5, line 240-243. Furthermore, we added 2 references (#73 page 6, line 266 and #77 page 6 line 282). In general, the molecular effects, mechanism of action of pyruvate, oxaloacetate and other small molecules in NVU cells require further studies.

Reviewer # 1:

  1. In paragraph 4, PLA, PGA, PCL, and PLGA were demonstrated as biodegradable polymers used to fabricate a scaffold that supports appropriate cell metabolism. These are all synthetic polymers, but it is necessary to summarize examples using natural polymers such as collagen/gelatin, alginate, and fibrin.

Authors: We added some information about natural polymers (page 12, line 541-545).

Reviewer # 1:

  1. Representative images (Figures from references or the authors’ own) should be added to each section.

Authors: According to the Reviewer’s suggestion, we added the figure 1 in section 2 (page 6) «Lactate/pyruvate and oxaloacetate/malate as regulators of NAD+/NADH ratio in mammalian cells», and figure 2 in section 4 (page 13) «Putative effect of hydrolysis of PLA/PLGA bioscaffolds on cell physiology in vitro».

We are very grateful for all the Reviewers’ comments and suggestions that help us to improve to quality of the manuscript.

Reviewer 2 Report

Dear Editor, In this review, authors have been focus on some carbohydrate metabolites – lactate, pyruvate, oxaloacetate, malate - that greatly contribute to the regulation of cell-to-cell communications and metabolic plasticity of brain cells, and on resorbable biopolymers that may reproduce the local environment enriched in particular cell metabolites. After carefully reading I found that the manuscript is well-written but there is also space for improvement. My specific comments are listed below.

The main problem of this review is that no one figure is presented. There are just 2 tables, which is unusual for review. So, I propose to add at least 4-5 figures with the main findings describing in this review. There are plenty of references and I am sure that authors will find some figures that can be uploaded free of charge.

Small molecules in the regulation of cell metabolism: I propose to add their chemical structures. Which is the mechanism of their activation-of their work? Please add some figures.

The same can be done in Features of neurovascular unit cell metabolism and local microenvironment in the regulation of brain plasticity. Which is the acting mechanism? Please add some figures.

Moderate editing of English language required

Author Response

Reviewer # 2:

Dear Editor, In this review, authors have been focus on some carbohydrate metabolites – lactate, pyruvate, oxaloacetate, malate - that greatly contribute to the regulation of cell-to-cell communications and metabolic plasticity of brain cells, and on resorbable biopolymers that may reproduce the local environment enriched in particular cell metabolites. After carefully reading I found that the manuscript is well-written but there is also space for improvement. My specific comments are listed below.

 Authors: Dear Reviewer! Thank you for the review of our manuscript. According to the Reviewer’s suggestion, we revised our article (all the corrections are indicated with yellow). 

Reviewer # 2:

The main problem of this review is that no one figure is presented. There are just 2 tables, which is unusual for review. So, I propose to add at least 4-5 figures with the main findings describing in this review. There are plenty of references and I am sure that authors will find some figures that can be uploaded free of charge.

Small molecules in the regulation of cell metabolism: I propose to add their chemical structures. Which is the mechanism of their activation-of their work? Please add some figures.

 The same can be done in Features of neurovascular unit cell metabolism and local microenvironment in the regulation of brain plasticity. Which is the acting mechanism? Please add some figures.

 Authors: According to the Reviewer’s suggestion, we provided figures to the manuscript to illustrate the chemical structures of small molecules, the mechanism of their activation and functioning in NVU.  Figure 1 in section 2 (page 6) «Lactate/pyruvate and oxaloacetate/malate as regulators of NAD+/NADH ratio in mammalian cells». Figure 2 in section 4 (page 13) «Putative effect of hydrolysis of PLA/PLGA bioscaffolds on cell physiology in vitro». Furthermore, we added a schematic image to the manuscript (in the attached file).

Reviewer # 2:

Moderate editing of English language required.

Authors: We edited the English language throw out the manuscript.

We are very grateful to the Reviewer for suggestions and additions that helped us to improve the presentation of throughout the manuscript.

Round 2

Reviewer 1 Report

The authors clearly and sufficiently provided replies to some critical issues raised in the previous review stage. It is considered that the present version of the manuscript was well revised according to all the reviewers' comments.
Thus, this manuscript would be acceptable, unless otherwise decided by other reviewers.

none